# ANODEV2: A Coupled Neural ODE Framework

**Tianjun Zhang**[1][*] **Zhewei Yao**[1][*] **Amir Gholami**[1][*]
**Kurt Keutzer**[1] **Joseph Gonzalez**[1] **George Biros**[2] **Michael W. Mahoney**[1,3]
[1]University of California at Berkeley, [2]University of Texas at Austin, [3]ICSI
{tianjunz, zheweiy, amirgh, keutzer, jegonzal, and mahoneymw}@berkeley.edu, biros@ices.utexas.edu

## Abstract

It has been observed that residual networks can be viewed as the explicit Euler discretization of an Ordinary Differential Equation (ODE). This observation motivated the introduction of so-called Neural ODEs, which allow more general discretization schemes with adaptive time stepping. Here, we propose ANODEV2, which is an extension of this approach that allows evolution of the neural network parameters, in a coupled ODE-based formulation. The Neural ODE method introduced earlier is in fact a special case of this new framework. We present the formulation of ANODEV2, derive optimality conditions, and implement the coupled framework in PyTorch. We present empirical results using several different configurations of ANODEV2, testing them on multiple models on CIFAR-10. We report results showing that this coupled ODE-based framework is indeed trainable, and that it achieves higher accuracy, as compared to the baseline models as well as the recently-proposed Neural ODE approach.

## 1 Introduction

Residual networks [1, 2] have enabled training of very deep neural networks (DNNs). Recent work has shown an interesting connection between residual blocks and ODEs, showing that a residual network can be viewed as a discretization to a continuous ODE operator [3, 4, 5, 6, 7, 8]. These formulations are commonly called *Neural ODEs* and here we follow the same convention. Neural ODEs provide a general framework that connects discrete DNNs to continuous dynamical systems theory as well as discretization and optimal control of ODEs, all subjects with very rich theory. A basic Neural ODE formulation and its connection to residual networks (for a single block in a network) is the following:

$$z_1 = z_0 + f(z_0, \theta) \qquad\qquad \text{ResNet,} \qquad\qquad (1a)$$

$$z(1) = z(0) + \int_0^1 f(z(t), \theta)dt \qquad\qquad \text{ODE,} \qquad\qquad (1b)$$

$$z(1) = z(0) + f(z_0, \theta) \qquad\qquad \text{ODE forward Euler.} \qquad (1c)$$

Here, $z_0$ is the input to the network and $z_1$ is the output activation; $\theta$ is the vector of network weights (independent of time); and $f(z, \theta)$ is the nonlinear operator defined by this block. (Here we have written the ODE $dz/dt = f(z, \theta)$ in terms of its solution at $t = 1$.) We can see that a single-step of forward Euler discretization of the ODE is identical to a traditional residual block. Alternatively, we could use a different time-stepping scheme or, more interestingly, use more time steps. Once the connection to ODEs was identified, several groups have incorporated the Neural ODE structure in neural networks and evaluated their performance on several different learning tasks.

A major challenge with training Neural ODEs is that backpropagating through ODE layers requires storage of all the intermediate activations (i.e., $z$) in time. In principle, the memory footprint of

---

[*]Equal contribution.

ODE layers has a cost of $\mathcal{O}(N_t)$ ($N_t$ is the number of time steps to solve the ODE layer), which is prohibitive. The recent work of [8] proposed an adjoint based method, with a training strategy that required only storage of the activation at the end of the ODE layer. All the intermediate activations were then "re-computed" by solving the ODE layers backwards. However, it has been recently shown that such an approach could lead to incorrect gradients, due both to numerical instability and also to inconsistencies that relate to optimizing infinite dimensional operators (the so called Discretize-Then-Optimize vs. Optimize-Then-Discretize issue) [9]. Moreover, importantly, it was observed that using other discretization schemes such as RK2 or RK4, or using more time steps, does not affect the generalization performance of the model (even with the DTO approach).

In this paper, building on the latter approach of [9], we propose ANODEV2, a more general Neural ODE framework that addresses this problem. ANODEV2 allows the evolution of *both weights and activations* by a coupled system of ODEs:

$$\begin{cases} z(1) = z(0) + \int_0^1 f(z(t), \theta(t))dt & \text{"parent network",} \\ \theta(t) = \theta(0) + \int_0^t q(\theta(t), p)dt, & \theta(0) = \theta_0 \ \text{"weight network".} \end{cases} \tag{2}$$

Here, $q(\cdot)$ is a nonlinear operator (essentially controlling the dynamics of the network parameters in time); $\theta_0$ and $p$ are the corresponding parameters for the weight network. Our approach allows $\theta$ to be time dependent: $\theta(t)$ is parameterized by the learnable dynamics of $d\theta/dt = q(\theta(t), p)$. This, in turn, is parameterized by $\theta_0$ and $p$. In other words, instead of optimizing for a constant $\theta$, we optimize for $\theta_0$ and $p$. During inference, *both* weights $\theta(t)$ and activations $z(t)$ are forward-propagated in time by solving Eq. 2. Observe that if we set $q = 0$ then we recover the Neural ODE approach proposed by [8]. Eq. 2 replaces the problem of designing appropriate neural network blocks ($f$) with the problem of choosing appropriate function ($q$) in an ODE to model the changes of parameter $\theta$ (the weight network).

In summary, our main contributions are the following.

- We provide a general framework that extends Neural ODEs to system of coupled ODEs which allows coupled evolution of both model parameters and activations. This coupled formulation addresses the challenge with Neural ODEs, in that using more time steps or different discretization schemes do not affect model's generalization performance [9].

- We derive the optimality conditions for how backpropagation should be performed for the coupled ODE formulation using by imposing the standard Karush–Kuhn–Tucker conditions. In particular, we implement the corresponding Discretize-Then-Optimize (DTO) approach, along with a checkpointing scheme presented in [9].

- We test the framework using multiple different residual models on Cifar-10 by considering different coupled formulations. In particular, we show examples illustrating how a biologically motivated reaction-diffusion-advection ODE could be used to model the evolution of the neural network parameters.

- We have open sourced the implementation of the coupled framework in Pytorch which allows general evolution operators (and not just the reaction-diffusion-advection). In fact, some of the earlier works such as HyperNetworks are special cases of ANODEV2, and they can be implemented in this framework. The code is available in [10].

There is a rich literature on neural evolution research [11, 12, 13, 14, 15, 16, 17]. Several similar approaches to ours have been taken in the line of evolutionary computing, where an auxiliary "child" network is used to generate the parameters for a "parent" network. This approach permits the restriction of the effective depth that the activations must go through, since the parent network could have smaller weight space than the child network. One example is HyperNEAT [18], which uses "Compositional Pattern Producing Networks" (CPPNs) to evolve the model parameters [19, 20]. A similar approach using "Compressed Weight Search" was proposed in [21]. A follow up work extended this approach by using differentiable CPPNs [22]. The authors show that neural network parameters could be encoded through a fully connected architecture. Another seminal work in this direction is [23, 24], where an auxiliary network learns to produce "context-aware" weights in a recurrent neural network model. A similar recent approach is taken in Hypernetworks [25]. In this approach, the model parameters are evolved through an auxiliary learnable neural network. This approach is a special case of the above framework, which could be derived by using a single time step

discretization of Eq. 2, with a neural network for the evolution operator (denoted by $q$ and introduced in the next section). Our framework is a generalization of these evolutionary algorithms, and it provides more flexibility for modeling the evolution of the model parameters in time. For instance, we will show how biologically motivated diffusion-reaction-advection operators could be used for the evolution operator $q$, with negligible increase in the model parameter size.

## 2 Methodology

In this section, we discuss the formulation for the coupled ODE-based neural network model described above, and we derive the corresponding optimality conditions. For a typical learning problem, the goal is to minimize the empirical risk over a set of training examples. Given a loss function $\ell_i$, where $i$ indexes the training sample, we seek to find weights, $\theta \in \mathbb{R}^d$, such that:

$$\min_{\theta} \frac{1}{N} \sum_{i=1}^{N} \ell_i(z_i(\theta)) + R(\theta), \tag{3}$$

where $R$ is a regularization operator and $N$ the number of training samples. The loss function depends implicitly on $\theta$ through the network activation vector $z_i$. This problem is typically solved using Stochastic Gradient Descent (SGD) and backpropagation to compute the gradient of $z_i$ with respect to $\theta$.

### 2.1 Neural ODE

Consider the following notation for a residual block: $z_1 = z_0 + f(z_0; \theta)$, where $z_0$ is the input activation, $f(\cdot)$ is the neural network kernel (e.g., comprising a series of convolutional blocks with non-linear or linear activation functions), and $z_1$ is the output activation. As discussed above, an alternative view of a residual network is the following continuous-time formulation: $\frac{dz}{dt} = f(z(t); \theta)$, with $z(t=0) = z_0$ and $z(t=1) = z_1$ (we will use both $z(t)$ and $z_t$ to denote activation at time $t$). In the ODE-based formulation, this neural network has a continuous depth. In this case, we need to solve the following constrained optimization problem (Neural ODE):

$$\min_{\theta} \frac{1}{N} \sum_{i=1}^{N} l_i(z_i(1)) + R(\theta) \quad \text{subject to:} \quad \frac{dz}{dt} = f(z(t), \ \theta), \quad z(0) = z_0. \tag{4}$$

Note that in this formulation the neural network parameters are stale in time. In fact it has been observed that using adaptive time stepping or higher order discretization methods such as Runge-Kutta does not result in any gains in generalization performance using the above framework [9]. To address this, we extend the Neural ODEs by considering a system of coupled ODEs, where the model parameters as well as activations evolve in time. In fact, this formulation is slightly more general than what we described in the introduction. For this reason, we introduce an auxiliary dynamical system for $w(t)$, which we use to define $\theta$. In particular, we propose the following formulation:

$$\min_{p,w_0} \mathcal{J}(z(1)) = \frac{1}{N} \sum_{i=1}^{N} l_i(z_i(1)) + R(w_0, p), \tag{5a}$$

$$\frac{dz}{dt} = f(z(t), \theta(t)), \ z(0) = z_0 \quad \text{``Activation ODE''}, \tag{5b}$$

$$\frac{\partial w}{\partial t} = q(w; p), \ w(0) = w_0 \quad \text{``Evolution ODE''}, \tag{5c}$$

$$\theta(t) = \int_0^t K(t - \tau) w(\tau) d\tau. \tag{5d}$$

Note that here $\theta(t)$ is a function of time, and it is parameterized by the whole dynamics of $w(t)$ and a time convolution kernel $K$ (which in the simplest form could be a Dirac delta function such that $\theta(t) = w(t)$). Also, $q(w, p)$ can be a general function, e.g., another neural network, a linear operator or even a discretized Partial Differential Equation (PDE) based operator. The latter perhaps is useful if we consider the $\theta(t)$ as a function $\theta(u, t)$, where $u$ parameterizes the signal space (e.g., 2D pixel space for images). This formulation allows for rich variations of $\theta(t)$, while using a lower

dimensional parameterization: notice that implicitly we have that $\theta(t) = \theta(w_0, p, t)$. Also, this formulation permits novel regularization techniques. Instead of regularizing $\theta(t)$, we can regularize $w_0$ and $p$.

A crucial question is: how should one perform backpropagation for this formulation? It is instructive to compute the actual derivatives to illustrate the structure of the problem. To derive the optimality conditions for this constrained problem, we need to first form the Lagrangian operator and derive the so called Karush–Kuhn–Tucker (KKT) conditions:

$$
\begin{aligned}
\mathcal{L} = \mathcal{J}(z(1)) &+ \int_0^1 \alpha(t) \cdot \left( \frac{dz}{dt} - f(z(t), \theta(t)) \right) dt + \int_0^1 \beta(t) \cdot \left( \frac{\partial w}{\partial t} - q(w; p) \right) dt \\
&+ \int_0^1 \gamma(t) \cdot \left( \theta(t) - \int_0^t K(t - \tau) w(\tau) d\tau \right) dt.
\end{aligned}
\tag{6}
$$

Here, $\alpha(t)$, $\beta(t)$, and $\gamma(t)$ are the corresponding adjoint variables (Lagrange multiplier vector functions) for the constraints in Eq. 5. The solution to the optimization problem of Eq. 5 could be found by computing the stationary points of the Lagrangian (KKT conditions), which are the gradient of $\mathcal{L}$ with respect to $z(t), w(t), \theta(t), p, w_0$ and the adjoints $\alpha(t), \beta(t), \gamma(t)$. The variations of $\mathcal{L}$ with respect to the three adjoint functions just result in the ODE constraints in Eq. 5. The remaining variations of $\mathcal{L}$ are the most interesting and are given below (see Appendix D for additional discussion on the derivation):

$$
\frac{\partial \mathcal{J}(z(1))}{\partial z_1} + \alpha_1 = 0, \quad -\frac{\partial \alpha}{\partial t} - \left( \frac{\partial f}{\partial z} \right)^T \alpha(t) = 0; \qquad (\partial \mathcal{L}_z) \quad \text{(7a)}
$$

$$
-\left( \frac{\partial f}{\partial \theta} \right)^T \alpha(t) + \gamma(t) = 0; \qquad (\partial \mathcal{L}_\theta) \quad \text{(7b)}
$$

$$
-\frac{\partial \beta(t)}{\partial t} - \left( \frac{\partial q}{\partial w} \right)^T \beta(t) - (1 - H(t)) \int_0^1 K^T(\tau - t) \gamma(\tau) d\tau = 0, \;\; \beta(1) = 0; \qquad (\partial \mathcal{L}_w) \quad \text{(7c)}
$$

$$
-\beta(0) + \frac{\partial R}{\partial w_0} = g_{w_0}; \qquad (\partial \mathcal{L}_{w_0}) \quad \text{(7d)}
$$

$$
\frac{\partial R}{\partial p} - \int_0^1 \left( \frac{\partial q}{\partial p} \right)^T \beta(t) dt = g_p; \qquad (\partial \mathcal{L}_p) \quad \text{(7e)}
$$

where $H(t)$ is the scalar Heaviside function. To compute the gradients $g_p$ and $g_{w_0}$, we proceed as follows. Given $w_0$ and $p$, we forward propagate $w_0$ to compute $w(t)$ and then $\theta(t)$. Then using $\theta(t)$ we can compute the activations $z(t)$. Then we solve the adjoint equations for $\alpha(t), \gamma(t)$ and $\beta(t)$, *in this order* Eq. 7a- 7e. Finally, the gradients of the loss function with respect to $p$ ($g_p$) and $w_0$ ($g_{w_0}$) are given from the last two equations. Notice that if we set $q = 0$ we will derive the optimality conditions for the Neural ODE without any dynamics for the model parameters, which was the model presented in [8]. The benefit of this more general framework is that we can encapsulate time dynamics of the model parameter without increasing the memory footprint of the model. In fact, this approach only requires storing initial condition for the parameters, which is parameterized by $w_0$, along with the parameters of the control operator $q$ which are denoted by $p$. As we show in the results section, the latter can have negligible memory footprint, but yet allow rich representation of model parameter dynamics.

**PDE-inspired formulation.** There are several different models for the $q(w, p)$, the evolution function for the weight convolutional network. One possibility is to use a convolutional block (resembling a recurrent network). However, this can increase the number of parameters significantly. Inspired by Turing's reaction-diffusion partial differential equation models for pattern formation, we view a convolutional filter as a time-varying pattern (where time here represents the depth of the network) [12]. To illustrate this, we consider a PDE based model for the control operator $q$, as follows:

$$
\frac{dw}{dt} = \sigma(\tau \Delta w + \upsilon \cdot \nabla w + \rho w),
\tag{8}
$$

where $\tau$ is used to control the diffusion ($\Delta w$), $\upsilon$ is used to control the advection ($\nabla w$), $\rho$ is used to control the reaction ($w$), and $\sigma$ is a nonlinear activation (such as sigmoid or tanh). View the weights

$w$ as a time series signal, starting from the initial signal, $w(0)$, and evolving in time to produce $w(1)$. In fact one can show that the above formulation can evolve the parameters to any weights, if there exists a diffeomorphic transformation of between the two distributions (i.e. if there exists a velocity field $\upsilon$ such $w(1)$ is the solution of Eq. 8, with initial condition $w(0)$ [26]). Although this operator is mainly used as an example control block (i.e., ANODEV2 is not limited to this model), but diffusion-reaction-advection operator can capture interesting dynamics for model parameters. For instance, consider a single Gaussian operator for a convolutional kernel, which is centered in the middle with a unit variance. A diffusion operator can simulate multiple different normal distributions with different variance in time. Note that this requires storing only a single diffusion parameter (i.e., $\tau$). Another interesting operator is the advection operator which models species transport. For the Gaussian case, this operator could for instance transport the center of the Gaussian to different positions other than the center of the convolution. Finally, the reaction operator, could allow growth or decay of the intensity of the convolution filters. The full diffusion-reaction-advection operator could encapsulate more complex dynamics of the neural network parameters in time. A synthetic example is shown in Figure 3 in the appendix, and a real example ($5 \times 5$ convolutional kernel of AlexNet) is shown in Figure 6 in the appendix.

## 2.2 Two methods used in this paper

We use two different coupling configurations of ANODEV2 as described below.

**Configuration One.** We use multiple time steps to solve for both $z$ and $\theta$ in the network instead of just one time step as in the original ResNet. Then the discretized solution of Eq. 10 in appendix will be as follows:

$$z_{t_0+\delta t} = z_{t_0} + \delta t f(z_{t_0}; \theta_{t_0}); \quad \theta_{t_0+\delta t} = \sigma \left( F^{-1} \left( \exp((-\tau k^2 + ik\upsilon + \rho)\delta t) F(\theta_{t_0}) \right) \right), \quad (9)$$

where $\delta t$ is the discretization time scale, and F is Fast Fourier Transform (FFT) operator (for the derivation, see the appendix). In this setting, we will alternatively update the value of $z$ and $\theta$ according to Eq. 9. Hence, the computational cost for an ODE block will be roughly $N_t$ times more expensive compared to that for the original residual block (same as in [8]). This network can be viewed as applying $N_t$ different residual blocks in the network but with neural network weights that evolve in time. Note that this configuration does not increase the parameter size of the original network except slight overhead of $\tau$, $\upsilon$ and $\rho$.

The first configuration is shown in top of Figure 1, where the model parameters and activations are solved with the same discretization. This is similar to the Neural ODE framework of [8], except that the model parameters are evolved in time for subsequent times, whereas in [8] the same model parameters are applied to the activations. [2] The dynamics of the model parameters are illustrated by different colors used for the convolution kernels in top of Figure 1. This configuration is equivalent to using the Dirac delta function for the $K$ function in Eq. 5d.

**Configurations Two.** ANODEV2 supports different coupling configurations between the dynamics of activations and model parameters. For example, it is possible to not restrict the dynamics of $\theta$ and $z$ to align in time, which is the second configuration that we consider. Here, we allow model parameters to evolve and only apply to activations after a fixed number of time steps. For instance, consider the Gaussian example illustrated in Figure 3. In configuration one, a residual block is created for each of the three time steps. However, in configuration two, we only apply the first and last time evolutions of the parameters (i.e., we only use $w_0$ and $w_1$ to apply to activations). This configuration allows sufficient time for the model parameters to evolve, and importantly limits the depth of the network that activations go through (see the bottom of Figure 1). In this case, the depth of the network is increased by a factor of two, instead of $N_t$, as in the first configuration (which is the approach used in [8, 9]). Both configurations are supported in ANODEV2, and we present preliminary results for both settings.

## 3 Results

In this section, we report the results of ANODEV2 for the two configurations discussed in section 2, on CIFAR-10 dataset which consists of 60,000 $32 \times 32$ colour images in 10 classes. The framework is

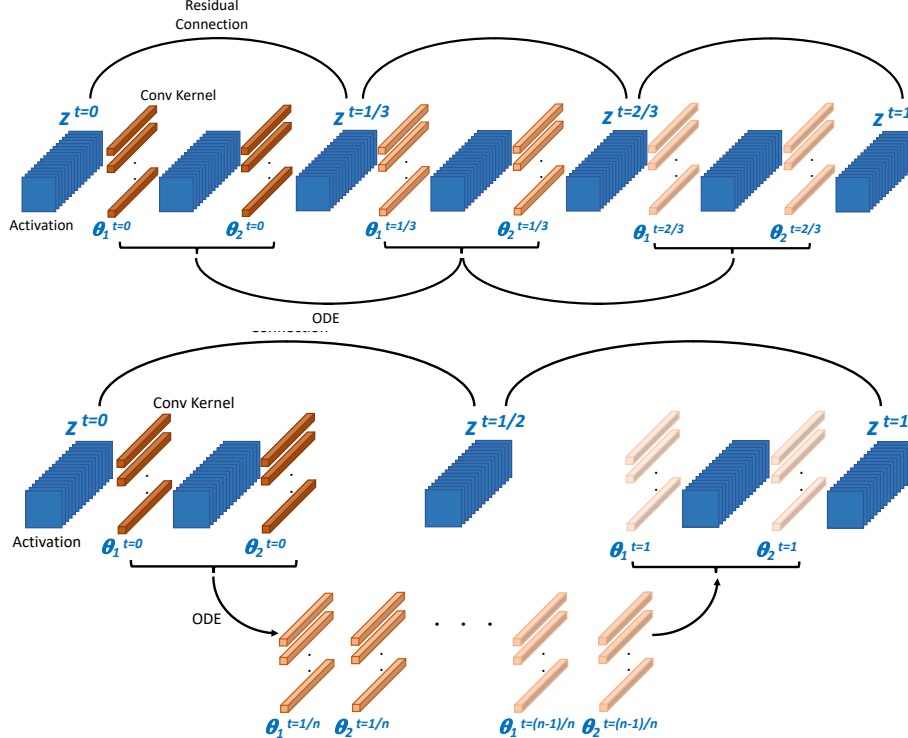

**Figure 1:** *Illustration of different configurations in* ANODEV2. *The top figure shows configuration one, where both the activation and weights θ are evolved through a coupled system of ODEs. During inference, we solve both of these ODEs forward in time. Blue squares in the figure represent activation with multiple channels; the orange bars represent the convolution kernel. The convolution weights θ are computed by solving an auxiliary ODE. The bottom figure shows configuration two, where first the weights are evolved in time before applying them to the activations.*

developed as a library in PyTorch and uses the checkpointing method proposed in [9], along with the discretize-then-optimize formulation of the optimality conditions shown in Eq. 7.

We test ANODEV2 on AlexNet with residual connection, as well as two different ResNets. See Appendix B and Appendix A.1 for the details of model architectures and training settings. We consider the two coupling configurations between the evolution of the activations and model parameters as discussed next.

## 3.1   Configuration One

We first start with configuration one. In this configuration, each time step corresponds to adding a new residual block in the network, as illustrated in Figure 1 (top). The results shown in Table 1. All the experiments were repeated five times, and we report both the min/max accuracy as well as the average of these five runs.

Note that the coupled ODE based approach outperforms the baseline in all of the three statistical properties above (i.e., min/max/average accuracy). For example, on ResNet-10 the coupled ODE network achieves 89.04% average test accuracy, as compared to 88.10% of baseline, which is 0.94% better. Meanwhile, a noticeable observation is that the minimum performance of the coupled ODE based network is comparable or even better than the maximum performance of baseline. The coupled ODE based AlexNet has 88.59% minimum accuracy, which is 1.44% higher than the best performance of baseline out of five runs. Hence, the generalization performances of the coupled ODE based network are consistently better than those of the baseline. It is important to note that the model parameter size of the coupled ODE approach in ANODEV2 is the same as that of the baseline. This is because the size of the control parameters $p$ is negligible. A comparison discussion is shown in section 4.1.

**Table 1:** *Results for using $Nt = 5$ time steps to solve $z$ and $\theta$ in neural network with configuration one. We tested on AlexNet, ResNet-4, and ResNet-10. We get 1.75%, 1.16% and 0.94% improvement over the baseline respectively. Note that the model size of the* ANODEV2 *and baseline is comparable.*

| | AlexNet | | ResNet-4 | | ResNet-10 | |
|---|---|---|---|---|---|---|
| | Min / Max | Avg | Min / Max | Avg | Min / Max | Avg |
| Baseline | 86.84% / 87.15% | 87.03% | 76.47% / 77.35% | 76.95% | 87.79% / 88.52% | 88.10% |
| ANODEV2 | **88.59% / 88.96%** | **88.78%** | **77.27% / 78.58%** | **78.11%** | **88.67% / 89.39%** | **89.04%** |
| Imp. | 1.75% / 1.81% | 1.75% | 0.80% / 1.23% | 1.16% | 0.88% / 0.87% | 0.94% |

The dynamics of how the neural network parameters are evolved in time is illustrated in Figure 6, where we extract the first $5 \times 5$ convolution of AlexNet and show how it evolves in time. Here, $Time$ represents how long $\theta$ evolves in time, i.e., $Time = 0$ shows the result of $\theta(t = 0)$ and $Time = 1$ shows the result of $\theta(t = 1)$. It can be clearly seen that the coupled ODE based method encapsulates more complex dynamics of $\theta$ in time. Similar illustrations for ResNet-4 and ResNet-10 are shown in Figure 4 and 5 in the appendix.

### 3.2 Configuration Two

Here, we test the second configuration where the evolution of the parameters and the activations could have different time steps. This means the parameter is only applied after a certain number of time steps of evolution but not at every time step which was the case in the first configuration. This effectively reduces the depth of the network and the computational cost, and it allows sufficient time for the neurons to be evolved, instead of naively applying them at each time step. An illustration for this configuration is shown in Figure 1 (bottom). The results on AlexNet, ResNet4 and ResNet10 are shown in Table 2, where we again report the min/max and average accuracy over five runs. As in the previous setting (configuration one), the coupled ODE based network performs better in all cases. The minimum performance of the coupled ODE based network still is comparable or even better than the maximum performance of the baseline. Although the overall performance of this setting is slightly worse than the previous configuration, the computational cost is much less, due to the smaller effective depth of the network that the activations go through.

**Table 2:** *Results for using $Nt = 2$ time steps to solve $z$ in neural network and $Nt = 10$ to solve $\theta$ in the ODE block (configuration two).* ANODEV2 *achieves* 1.23%, 0.78% and 0.83% *improvement over the baseline respectively. Note that the model size is comparable to baseline Table 1.*

| | AlexNet | | ResNet-4 | | ResNet-10 | |
|---|---|---|---|---|---|---|
| | Min / Max | Avg | Min / Max | Avg | Min / Max | Avg |
| Baseline | 86.84% / 87.15% | 87.03% | 76.47% / 77.35% | 76.95% | 87.79% / 88.52% | 88.10% |
| ANODEV2 | **88.1% / 88.33%** | **88.26%** | **77.23% / 78.28%** | **77.73%** | **88.65% / 89.19%** | **88.93%** |
| Imp. | 1.26% / 1.18% | 1.23% | 0.76% / 0.93% | 0.78% | 0.86% / 0.67% | 0.83% |

## 4  Ablation Study

Here we perform an ablation study in which we remove the evolution of the model parameters, and instead we fix them to stale values in time (which is the configuration used in [8, 9]), and test with a case where the model parameters are indeed evolved in time, which corresponds to results of Table 2. Precisely, we use two time steps for activation ODE (Eq. 5b) and ten time steps for the evolution of the model parameters (Eq. 5c). In this setting, both the FLOPS and model sizes are the same, allowing us to test the efficacy of evolving model parameters. The results are shown in Table 4. As one can see, there is indeed benefit in allowing the model parameter to evolve in time, which is rather obvious since it gives more flexibility to the neural network to evolve the model parameters. The ANODE results are derived using the DTO approach with checkpointing presented in [9]. We also tested Neural ODE approach used in [8], the results are significant worse than ANODE and ANODEV2. Also note that evolving model parameters has a negligible computational cost, since we can actually use analytical solutions for solving the reaction-diffusion-advection, which is discussed in Appendix A.1.

**Table 3:** *Parameter comparison for two* ANODEV2 *configurations, the network used in section 4, and the baseline network. The parameter size of* ANODEV2 *is comparable with others.*

|  | AlexNet | ResNet-4 | ResNet-10 |
|---|---|---|---|
| Baseline | 1756.68K | 7.71K | 44.19K |
| ANODEV2 config. 1 | 1757.51K | 8.23K | 45.77K |
| ANODEV2 config. 2 | 1757.13K | 7.99K | 45.05K |
| Neural ODE [8] / ANODE [9] | 1757.13K | 7.96K | 44.95K |

**Table 4:** *We use* $Nt = 2$ *time steps to solve* $z$ *in neural network and keep* $\theta$ *as static for Neural ODE and* ANODE. *We tested all the configurations on AlexNet, ResNet-4 and ResNet-10. The results shows that Neural ODE get significantly worse results comparing to* ANODEV2 *and* ANODE. ANODEV2 *get* 0.24%, 0.43% *and* 0.33% *improvement over* ANODE *respectively. The model size comparison is shown in Table 3.*

|  | AlexNet | | ResNet-4 | | ResNet-10 | |
|---|---|---|---|---|---|---|
|  | Min / Max | Avg | Min / Max | Avg | Min / Max | Avg |
| Baseline | 86.84% / 87.15% | 87.03% | 76.47% / 77.35% | 76.95% | 87.79% / 88.52% | 88.10% |
| NeuralODE [8] | 74.54% / 76.78% | 75.67% | 44.73% / 49.91% | 47.37% | 64.7% / 70.06% | 67.94% |
| ANODE [9] | 87.86% / 88.14% | 88.02% | 76.92% / 77.45% | 77.30% | 88.48% / 88.75% | 88.60% |
| ANODEV2 | **88.1% / 88.33%** | **88.26%** | **77.23% / 78.28%** | **77.73%** | **88.65% / 89.19%** | **88.93%** |

### 4.1 Parameter Size Comparison

Here, we provide the parameter sizes of the two configurations tested above and the model used in ablation study in section 4. It can be clearly seen that the model sizes of both configurations are roughly the same as those of the baseline models. In fact, configuration one grows the parameter sizes of AlexNet, ResNet-4, and ResNet-10 by only $0.5\%$ to $6.7\%$, as compared to those of baseline models. In configuration two, the parameter size increases from $0.2\%$ to $3.6\%$ compared to baseline model (note that we even count the additional batch norm parameters for fair comparison). Comparing with the ablation network used in section 4, in which we apply the same model parameters for multiple time steps, ANODEV2 configuration two has basically the same number of parameters. Table 3 summarizes all the results.

## 5 Conclusions

The connection between residual networks and ODEs has been recently found in several works. Here, we propose ANODEV2, which is a more general extension of this approach by introducing a coupled ODE based framework, motivated by the works in neural evolution. The framework allows dynamical evolution of both the residual parameters as well as the activations in a coupled formulation. This gives more flexibility to the neural network to adjust the parameters to achieve better generalization performance. We derived the optimality conditions for this coupled formulation and presented preliminary experiments using two different configurations, and we showed that we can indeed train such models using our differential framework. The results on three neural networks (AlexNet, ResNet-4, and ResNet-10) all showed accuracy gains across five different trials. In fact the worst accuracy of the coupled ODE formulation was better than the best performance of the baseline. This is achieved with negligible change in the model parameter size. To the best of the our knowledge, this is the first coupled ODE formulation that allows for the evolution of the model parameters in time along with the activations. We are working on extending the framework for other learning tasks. The source code will be released as open source software to the public.

# 6  Rebuttal

We would like to thank all the reviewers and area chair for taking the time to review our work and providing us with their valuable feedback.

Here we present solving the problem of simulating 1D wage equation with ANODEV2. For example, we can directly capture a variable velocity wave equation, using the reaction term in the evolution kernel. This is illustrated in Figure 2 below, where we test a simple transport phenomena with variable velocity. The governing equation here is dz/dt = c(t) dz/dx, where c(t) is variable velocity, and z is the signal that changes in time. The learning task is to predict how the signal changes in time. That is we are given z(t=0) and want to infer z(t) at different time points. We test a one layer model in 1D and illustrate the results in Figure 2 above. ANODEV2 can easily capture variable velocity (c(t)) with only a single layer through reaction operator, and as you can see the quality of its prediction is better than ANODE. We emphasize that this is a simple problem, and we are now investigating more complex physics based problems, for which we anticipate ANODEV2 to perform better by incorporating physical constraints in the evolution kernel.

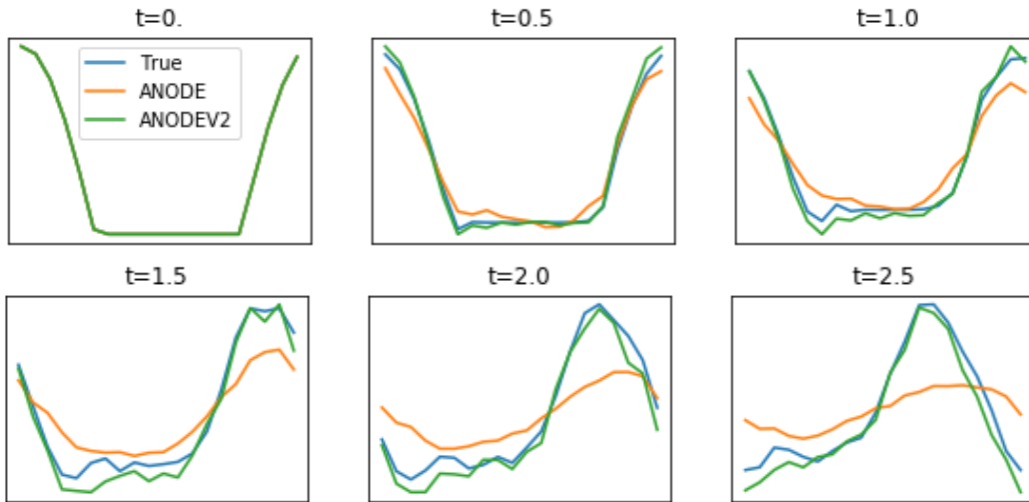

**Figure 2:** *Reconstruction of a signal transport problem. Here the task is to predict the change of the input signal (shown in the top left at $t = 0$) in time. The governing equation is a first-order wave equation with variable velocity in time. The blue curve shows ground truth, orange shows* ANODE, *and green shows* ANODEV2. *We used a single layer model to learn the transport equation.* ANODEV2 *performs better, as it can capture the transport physics as a constraint through the Turing's reaction operator. The x-axis shows spatial location, and y-axis is signal amplitude.*

## Acknowledgments

This work was supported by a gracious fund from Intel corporation, Berkeley Deep Drive (BDD), and Berkeley AI Research (BAIR) sponsors. We would like to thank the Intel VLAB team for providing us with access to their computing cluster. We also gratefully acknowledge the support of NVIDIA Corporation for their donation of two Titan Xp GPU used for this research. We would also like to acknowledge ARO, DARPA, NSF, and ONR for providing partial support of this work.

## Footnotes

[2]To be precise the possibility of using time varying parameters was discussed in [8] but the experiments were only limited to concatenating time to channels. Our approach here is more general, as it allows the NN parameters to evolve in time.

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
