[Supplementary Material]

# ANODEV2: A Coupled Neural ODE Framework

**Anonymous (Full Paper with Appendix)**

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

**Figure 1:** *Illustration of how different convolutional operators are evolved in time during the coupled neural ODE solve (through the evolution operator q). The figure corresponds to the first channel of the first convolution kernel parameters of AlexNet. These filters will be applied to activation in different time steps (through the f operator in the coupled formulation). As the time step increases, the kernel turns out to focus on some specific part on the activation map. Similar illustrations for ResNet-4 and ResNet-10 are shown in Figure 4 and 5 in the appendix.*

## 2.2 Two methods used in this paper

We use two different coupling configurations of ANODEV2 as described below.

**Configuration One.** We use multiple time steps to solve for both $z$ and $\theta$ in the network instead of just one time step as in the original ResNet. Then the discretized solution of Eq. 10 in appendix will be as follows:

$$z_{t_0+\delta t} = z_{t_0} + \delta t f(z_{t_0}; \theta_{t_0}); \quad \theta_{t_0+\delta t} = \sigma \left( F^{-1} \left( \exp((-\tau k^2 + ik\upsilon + \rho)\delta t) F(\theta_{t_0}) \right) \right). \quad (9)$$

where $\delta t$ is the discretization time scale, and F is Fast Fourier Transform (FFT) operator (for derivation please see appendix). In this setting, we will alternatively update the value of $z$ and $\theta$ according to Eq. 9. Hence, the computational cost for an ODE block will be roughly $N_t$ times more expensive compared to that for the original residual block (same as in [8]). This network can be viewed as applying $N_t$ different residual blocks in the network but with Neural Network weights that evolve in time. Note that this configuration does not increase the parameter size of the original network except slight overhead of $\tau$, $\upsilon$ and $\rho$.

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

## A    RDA Simulation

In this section, we provide the details for the reaction-diffusion-advection solver and an exemplary simulation shown in Figure 3. For illustration of the idea, we set the initial distribution of $\theta$ to be a unit Gaussian centered in the middle. In the first row, we show how this single modal Gaussian changes in time when only diffusion operator is used in the control operator. As shown in the figure, the diffusion operator allows the parameters to evolve from a Gaussian with unit variance, to Gaussian filters with higher variance. A similar illustration is shown in the second row with advection operator. Notice how the advection operator allows modeling of different filters centered at different locations with the same variance (since advection operator does not diffuse filters but transports them). The third row shows the simulation when we only use an exponential growth operator for the reaction part. Notice how this operator could allow the kernel to increase/decrease its intensities at different pixels in time. Finally in the last row, we show an example where we use all three operators together.

**Figure 3:** *Illustration of how different convolution maps could be encoded through the parameter PDE solver. Here we show an exemplary convolution at time $t = 0$ (left image), as well as its evolution through time, when we apply the reaction-diffusion-advection (RDA) PDE for the model parameters. Note that with this PDE based encoding, we only need to store the initial condition for the parameters (i.e., $t = 0$). The rest of the model parameters could be computed using this initial condition.*

### A.1    Numerical Method

We set $K$ to be a Dirac delta function, and use the above reaction-diffusion-advection function for $q(w, p)$. We have

$$\begin{cases} \frac{dz}{dt} = f(z; \theta), \\ \frac{dw}{dt} = \sigma(\tau \Delta w + \upsilon \cdot \nabla w + \rho w). \end{cases} \quad (10)$$

Here, we discuss how can we solve the ODE system Eq. 10. For the evolution of $z$, we follow [9] and use forward Euler method to solve $z$. For example, if we set time step, $N_t$ to be 2, then

$$z_{1/2} = z_0 + \frac{1}{2}f(z_0; \theta_0); \quad z_1 = z_{1/2} + \frac{1}{2}f(z_{1/2}; \theta_{1/2}).$$

It is not hard to see that if $N_t = 1$, then the output is the same as the original ResNet. For the evolution of $\theta$ without non-linearity, i.e. $\sigma$ is the identical map, there exists an analytic solution can in the frequency domain. Applying Fast Fourier Transform (FFT) from Eq. 8 we will get:

$$F(w)_t = F(\tau\Delta w + \upsilon \cdot \nabla w + \rho w), \tag{11}$$

where $F(\cdot)$ denotes FFT operator. Since the diffusion, advection, and reaction coefficients are constant, we can find the analytical solution in the frequency domain. That is:

$$w_{t_0 + \delta t} = F^{-1}\big(\exp(-\delta t\tau k^2 + ik\delta t\upsilon + \delta t\rho)F(w_{t_0})\big), \tag{12}$$

where $F^{-1}$ is inverse FFT. Note that due to the existence of this analytical solution the computational cost of solving the evolution for $\theta$ becomes negligible which is an important benefit of this approach. When non-linearity is applied, we use an approximation to solve Eq. 8,

$$w_{t_0 + \delta t} = \sigma\left(F^{-1}\left(\exp(-\delta t\tau k^2 + ik\delta t\upsilon + \delta t\rho)F(w_{t_0})\right)\right). \tag{13}$$

This means we first apply FFT and its inverse to solve the linear system then apply the non-linear function $\sigma$. Here, $\delta t$ means the time scale to compute $\theta$. Also, in this paper we set the non-linearity function $\sigma$ to be tanh. However, other non-linearities could also be used. For configuration 1, we use $N_t = 5$. And for configuration 2, we use $N_t = 2$ to solve $z$ and $N_t = 10$ to solve $\theta$. In this configuration, the FLOPS will be only $2\times$ of the original baseline network. Upon this condition, the process can be formulated as,

$$z_{1/2} = z_0 + \frac{1}{2}f(z_0; \theta_0); \quad z_1 = z_{1/2} + \frac{1}{2}f(z_{1/2}; \theta_1);$$

where $\theta_1$ is generated with $\delta t = 1/10$.

## B    Model Configuration

In this section, we provide the architecture we used for the tests in section 3. The AlexNet, ResNet-4 and ResNet-10 we are using are described in following sections.

### B.1    AlexNet

We used a 2-layer convolution with residual connection added to the second convolution. Thus, we can transform the second convolution into an ODE. Table 5 explains detailed structure layer by layer. For simplicity, we omit the batch normalization and ReLU layer added after each convolution.

**Training details.**    We train AlexNet for 120 epochs with initial learning rate 0.1. The learning rate decays by a factor of 10 at epoch 40, 80 and 100. Data augmentation is implemented. Also, the batch size used for training is 256. Note that the setting is the same for all experiments, i.e. baseline, Neural ODE, and ANODEV2.

### B.2    ResNet-4 and ResNet-10

Here, we provide the architecture of ResNet-4 and ResNet-10 used section 3. We also omit the batch normalization and ReLU for simplicity. Detailed structure are provided in Table 6.

**Training details.**    We train ResNet-4/10 for 350 epochs with initial learning rate 0.1. The learning rate decays by a factor of 10 at epoch 150, and 300. Data augmentation is implemented. Also, the batch size used for training is 256. Note that the setting is the same for all experiments, i.e. baseline, Neural ODE, and ANODEV2.

## C    Convolution kernel Evolution Example

In this section, we show some examples of how the model parameters $\theta$ are evolved in time. Results for ResNet-4 and ResNet-10 are shown in Figure 4 and Figure 5 respectively.

**Table 5:** *Summary of the architecture used in AlexNet. This is a 2-convolution network with residual connection added to the second convolution, followed by three fully connected layer.*

| Name | output size | Channel In / Out | Kernel Size | Residual |
|---|---|---|---|---|
| conv1 | 32×32 | 3 / 64 | 5×5 | No |
| max pool | 16×16 | 64 / 64 | - | - |
| conv2 | 16×16 | 64 / 64 | 5×5 | Yes |
| max pool | 8×8 | 64 / 64 | - | - |

| Name | input size | output size |
|---|---|---|
| fc1 | 4096 | 384 |
| fc2 | 384 | 192 |
| fc3 | 192 | 10 |

**Table 6:** *Summary of the architecture used in ResNet-4 and ResNet-10. ResNet-10 is a ResNet family that has 2 layers with 2 residual blocks in each layer. ResNet-4 has only 1 layer with only 1 residual block inside.*

| Name | output size | Channel In / Out | Kernel Size | Residual | Blocks(ResNet-4 / ResNet-10) |
|---|---|---|---|---|---|
| conv1 | 32×32 | 3 / 16 | 3×3 | No | 1 / 1 |
| layer1_1 | 32×32 | 16 / 16 | $\begin{bmatrix} 3\times3 \\ 3\times3 \end{bmatrix}$ | Yes | 1 / 1 |
| layer1_2 | 32×32 | 16 / 16 | $\begin{bmatrix} 3\times3 \\ 3\times3 \end{bmatrix}$ | Yes | 0 / 1 |
| layer2_1 | 16×16 | 16 / 32 | $\begin{bmatrix} 3\times3 \\ 3\times3 \end{bmatrix}$ | Yes | 0 / 1 |
| layer2_2 | 16×16 | 32 / 32 | $\begin{bmatrix} 3\times3 \\ 3\times3 \end{bmatrix}$ | Yes | 0 / 1 |

| Name | Kernel Size | Stride | Output Size (ResNet-4/ResNet-10) |
|---|---|---|---|
| max pool | 8×8 | 8 | 4×4 / 2×2 |

| Name | input size (ResNet-4/ResNet-10) | output size |
|---|---|---|
| fc | 256 / 128 | 10 |

**Figure 4:** *Illustration of how different convolution operators are evolved in time during the neural ODE solve. This is one channel of the first convolution in first layer in ResNet-4. Similar pattern can be observed as Figure 1.*

**Figure 5:** *Illustration of how different convolution operators are evolved in time during the neural ODE solve. This is one channel of the first convolution in first layer in ResNet-10. Similar pattern can be observed as Figure 1.*

## D  Derivation of Optimality Conditions

Here we present detailed derivation of the optimality conditions corresponding to Eq. 5. We need to find the so called KKT conditions, which can be found by finding stationary points of the corresponding Lagrangian, defined as:

$$
\begin{aligned}
\mathcal{L} &= \mathcal{J}(z_1) + \int_0^1 \alpha(t) \cdot \left( \frac{dz}{dt} - f(z(t), \theta(t)) \right) dt + \int_0^1 \beta(t) \cdot \left( \frac{\partial w}{\partial t} - q(w, p) \right) dt \\
&+ \int_0^1 \gamma(t) \cdot \left( \theta(t) - \int_0^t K(t - \tau) w(\tau) d\tau \right) dt + \alpha_0 \cdot (z_0 - z(0)) + \beta_0 \cdot (w_0 - w(0)).
\end{aligned}
\tag{14}
$$

In order to derive the optimality conditions, we first take variations with respect to $\alpha(t)$, $\beta(t)$, and $\gamma(t)$. This basically results in the "Activation ODE", the "Evolution ODE", and the relation between $\theta(t)$ and $w(t)$, shown in Eq. 5. Taking variations with respect to $z(t)$ will result in a backward-in-time ODE for the $\alpha(t)$, which is continuous equivalent to backpropagation. Taking variations with respect to $\theta$ will result in an algebraic relation between $\alpha(t)$ and $\gamma(t)$; taking variations with respect to $w(t)$ will be split in two parts. Variations with respect to $w(t)$ for $t > 0$; and with respect to $w(0)$ which is in fact one of our unknown parameters. The split is done by first integrating by parts the $\int_0^1 \beta(t) dw(t)/dt$ term to expose a term that reads $\beta(1)w(1) - \beta(0)w_0$, and then taking variations with respect to $w_0$. Finally, we also need to take variations with respect to the vector $p$. One small technical detail is that to take the variations of the $\int_0^1 \gamma(t) \cdot \int_0^t K(t - \tau) w(\tau) d\tau dt$ with respect to $w$ can be done easily by converting it $\int_0^1 \gamma(t) \cdot \int_0^1 (1 - H(t)) K(t - \tau) w(\tau) d\tau \, dt$. The details are given below.

In order to satisfy the first optimality condition on $z$ we have:

$$
(\frac{\partial \mathcal{L}}{\partial z})^T \hat{z} = 0,
$$

where this equality must hold for any variation $\hat{z}$ in space and time. We have:

$$(\frac{\partial \mathcal{L}}{\partial z})^T \hat{z} = \left(\frac{\partial \mathcal{J}(z_1)}{\partial z_1}\right)^T \hat{z}_1 + \int_0^1 \left(-\frac{\partial \alpha}{\partial t} - \frac{\partial f(z,\theta)}{\partial z}^T \alpha\right)^T \hat{z} dt + (\alpha_1^T \hat{z}_1 - \alpha_0^T \hat{z}_0) + \alpha_0^T \hat{z}_0$$

$$= (\frac{\partial \mathcal{J}(z_1)}{\partial z_1})^T \hat{z}_1 + \alpha_1^T \hat{z}_1 + \int_0^1 \left(-\frac{\partial \alpha}{\partial t} - \frac{\partial f(z,\theta)}{\partial z}^T \alpha\right)^T \hat{z} dt = 0. \tag{15}$$

Imposing this condition holds for all variation $\hat{z}$ will result in the first adjoint equation as follows:

$$\frac{\partial \mathcal{J}(z(1))}{\partial z_1} + \alpha_1 = 0, \quad -\frac{\partial \alpha}{\partial t} - \left(\frac{\partial f}{\partial z}\right)^T \alpha = 0. \tag{16}$$

For $\theta$, the following equation needs to be satisfied:

$$(\frac{\partial \mathcal{L}}{\partial \theta})^T \hat{\theta} = 0.$$

We have

$$(\frac{\partial \mathcal{L}}{\partial \theta})^T \hat{\theta} = \int_0^1 \left(-\frac{\partial f(z,\theta)}{\partial \theta}\right)^T \alpha^T \hat{\theta} dt + \int_0^1 \gamma^T \hat{\theta} dt. \tag{17}$$

This further implies:

$$-(\frac{\partial f(z,\theta)}{\partial \theta})^T \alpha + \gamma = 0. \tag{18}$$

Finally, the inversion equation on $w$ could be found by performing variation on $w$:

$$(\frac{\partial \mathcal{L}}{\partial w})^T \hat{w} = 0.$$

We have

$$(\frac{\partial \mathcal{L}}{\partial w})^T \hat{w} = \int_0^1 -(\frac{\partial \beta}{\partial t} - \frac{\partial q(w;p)}{\partial w}^T \beta)^T \hat{w} dt$$

$$+ \beta_1^T \hat{w}_1 + \int_0^1 (1 - H(t)) \int_0^t -(K^T(t-\tau)\gamma)^T d\tau \hat{w} dt$$

$$= \int_0^1 -(\frac{\partial \beta}{\partial t} - \frac{\partial q(w;p)}{\partial w}^T \beta)^T \hat{w} dt$$

$$+ \beta_1^T \hat{w}_1 + \int_0^1 (1 - H(t)) \int_0^t -(K^T(t-\tau)\gamma)^T d\tau \hat{w} dt, \tag{19}$$

where $H(t)$ is the scalar Heaviside function. Imposing this condition holds for all variation $\hat{w}$ will result in the inversion equation as follows,

$$-\frac{\partial \beta}{\partial t} - (\frac{\partial q(w;p)}{\partial w})^T \beta + (1 - H(t)) \int_0^t -K^T(t-\tau)\gamma d\tau, \quad \beta_1 = 0. \tag{20}$$

The gradient of $\mathcal{L}$ with respect to $w_0$ can be computed as,

$$g_{w_0} = \frac{\partial \mathcal{L}}{\partial w_0} = \frac{\partial R(w_0, p)}{\partial w_0} - \beta_0. \tag{21}$$

Finally, the gradient of $\mathcal{L}$ with respect to $p$ can be computed as,

$$g_p = \frac{\partial \mathcal{L}}{\partial p} = \frac{\partial R(w_0, p)}{\partial p} - \int_0^1 (\frac{\partial q(w, p)}{\partial p})^T \beta(t) dt. \tag{22}$$

Note that if optimality conditions are achieved with respect to $w_0$ and $p$, then

$$g_{w_0} = 0, \quad g_p = 0. \tag{23}$$