[Reviews · NeurIPS 2019]

Reviewer 1



[Originality] This paper is one of the first papers, if not the very first, to introduce a coupled ODE framework that takes a principled approach on neural ODE to dynamically controlling the neural net parameters. There exist some previous papers on dynamic parameters in neural nets, but they are not quite related to neural ODE and specifically not looking into the discretization issues in neural ODE. And with one additional coupled ODE, the presented framework can perform inference efficiently. [Quality] The theoretical part of this paper is sound and mostly self-contained. The authors presented detailed experiment results with several examples of the proposed framework. It clearly shows that the proposed coupled ODE framework has advantage over the original neural ODE. And I believe the authors makes a good job in evaluating their and others' work. [Clarity] The paper is clearly written with some visualizations for readers to understand the proposed framework. Key equations for the variants of coupled ODEs are provided. [Significance] The paper presented an efficient way to improve neural ODE via allowing a separate dynamic weight evolution ODE and coupling this one with the original ODE. The performance gain is likely to indicate that this approach is effective. And the use of dynamic parameters in neural ODE may inspire other work in different applications which involve both dynamical systems and requirement for adaptability (for example, robotics).

Reviewer 2



In Eqn 4 and Eqn 5, the z_0 should carry an index i for the i^th training example, i.e., the index shows up only in the loss l_i, but not in the dynamics. In the Neural ODE paper of [8] time dependence is explicitly included in the function f: dh/dt = f(h, theta, t) In the architecture, it enters through an extra channel for time. While the coupled ODE formulation of the paper is elegant, It is unclear whether this mechanism of introducing time-varying weights is better and if so precisely why. The paper grounds its motivation through the observations in [9]: "other discretization schemes such as RK2 or RK4, or using more time steps does not affect the generalization performance of the model". However, it is not made precisely clear how the ANODEv2 approach resolves these problems. The reference to Turings paper on The Chemical Basis of Morphogenesis has the wrong year. The diffusion-reaction-advection model for convolutional weights is interesting and worthwhile to study in greater detail. The Baseline in the experiments is not described. Overall the improvements over Neural ODE seem smallish but consistent. Which integrators are used in the implementation? Timings are not reported.

Reviewer 3



Strengths: 1. The PDE-inspired formulation of coupled ODE is very interesting and can enable utilization of decades of progress in efficiently solving particular classes of coupled equations, in deep learning applications. This is a very exciting connection discovered by the authors. 2. The general idea of allowing activations and weights to evolve (in particular, evolve independently) is an interesting approach to enrich neuralODE representation. Weaknesses: 1. The central contribution of modeling weight evolution using ODEs hinges on the mentioned problem of neural ODEs exhibiting inaccuracy while recomputing activations. It appears a previous paper first reported this issue. The reviewer is not convinced about this problem. The current paper doesn't provide a convincing analytical argument or empirical evidence about this issue. 2. Leaving aside the claimed weakness of neuralODE, the idea of modeling weight evolution as ODE is itself very intellectually interesting and worthy of pursuit. But the empirical improvement reported in Table 1 over AlexNet, ResNet-4 and ResNet-10 is <= 1.75 % for both configurations. The improvement of decoupling weight evolution is in fact even small and not consistent - the improvement in ResNet for configuration 2 is smaller than keeping the evolution of parameters and activations aligned. The improvement for ablation study over neuralODE is also minimal. So, the empirical case for the proposed approach is not convincing. 3. The derivation of optimality conditions for the coupled formulation is interesting because of connections to a machine learning application (backpropagation) but a pretty standard textbook derivation from dynamical systems / controls point of view.

[Author Response · NeurIPS 2019]

| | ResNet-4 | | ResNet-10 | | ResNet-14 | |
|---|---|---|---|---|---|---|
| | Min / Max | Avg | Min / Max | Avg | Min / Max | Avg |
| Baseline | 76.47%/77.35% | 76.95% | 87.79%/88.52% | 88.10% | 91.21%/92.12% | 91.68% |
| NeuralODE[8] | 52.58%/56.65% | 54.47% | 68.71%/70.48% | 69.43% | 75.32%/76.79% | 76.13% |
| ANODE | 77.13%/78.00% | 77.54% | 88.39%/88.87% | 88.70% | 91.66%/92.13% | 91.90% |
| **ANODEV2** | **77.55%/78.07%** | **77.83%** | **88.82%/89.19%** | **88.97%** | **92.13%/92.35%** | **92.19%** |

Table-R 1: We report results for Neural ODE[8], ANODE[9], and ANODEV2 (using configuration 2) for various models on Cifar-10. We report results where we concatenate time as an additional channel as requested by **R2**. All results were repeated five times with different random seeds. As one can see, ANODEV2 provides statistically consistent improvements. Also, for dynamical problems as shown in Fig-R 1, you can see a clear advantage of ANODEV2 as compared to ANODE[9] (and Neural ODE[8]).

Figure-R 1: Reconstruction of a signal transport problem. Here the task is to predict the change of the input signal (shown in the top left at $t = 0$) in time. The governing equation is a first-order wave equation with variable velocity in time. The blue curve shows ground truth, orange shows ANODE, and green shows ANODEV2. We used a single layer model to learn the transport equation. ANODEV2 performs better, as it can capture the transport physics as a constraint through the Turing's reaction operator. The x-axis shows spatial location, and y-axis is signal amplitude.

1. We thank all the reviewers for taking the time to review our work and provide their constructive feedback.

2. 1. **R1/R2:** Elaborate more on the advantage of designing the PDE in this specific way? and show effectiveness on learning dynamical systems, similar to Neural ODE paper. A: For vision based problems, Turing's reaction-diffusion model (with advection) can simulate several different filter shapes, as shown in Figs. 1 and 3 in the paper. This allows us to use the evolution kernel to capture specific features of the problem. For example, we can directly capture a variable velocity wave equation, using the reaction term in the evolution kernel. This is illustrated in Figure-R. 1 above, where we test a simple transport phenomena with variable velocity. The governing equation here is dz/dt = c(t) dz/dx, where c(t) is variable velocity, and z is the signal that changes in time. The learning task is to predict how the signal changes in time. That is we are given z(t=0) and want to infer z(t) at different time points. We test a one layer model in 1D and illustrate the results in Fig.-R 1 above. ANODEV2 can easily capture variable velocity (c(t)) with only a single layer through reaction operator, and as you can see the quality of its prediction is better than ANODE. We emphasize that this is a simple problem, and we are now investigating more complex physics based problems, for which we anticipate ANODEV2 to perform better by incorporating physical constraints in the evolution kernel. We will add this result in the paper. We should also note that the evolution kernel does not have to be based on Turing's system, and other kernels could be used depending on the target application.

2. **R2:** Unclear whether introducing time-varying weights is better than neural ODE paper which adds time through an extra channel. A: With the evolution kernel we can encapsulate different filters without having to store additional filters in memory, which is not possible through time concatenation as performed in [8]. This was illustrated in Figs. 1 and 3 in the paper. We have also added an ablation study in Tab-R 1 above which compares ANODEV2 with time concatenation as done in [8]. For fair comparison we have also included results with ANODE which addresses numerical instability of [8]. All the results were repeated five times and we report the statistics. ANODEV2 provides small but statistically consistent improvements (please note that we did not perform any hyper-parameter tuning).

3. **R2/R3:** Overall the improvements over Neural ODE seem smallish but consistent. A: That is true but the improvements are statistically consistent. We should mention that there are various factors that need to be studied, for instance initialization of the evolution kernel parameters, which can play an important role in the final generalization of the model. This is a non-convex optimization, and initialization as well as hyper-parameter selections can affect the results. Please note that this is the very first work in this area. see above ANODEV2 performs significantly better for dynamical systems, and this is also without tuning. We anticipate that having a NN model with evolutionary kernel could be very useful for physics based learning problems.

4. **R2:** (i) The Baseline description (ii) integrators used (iii) Timings report A: The baseline is basically a residual network which is equivalent to $N_t = 1$ without parameter evolution. We used Euler time stepping for our experiments but our code in [10] supports (Runge Kutta) RK2, RK4, and RK45 integrators. In terms of flops/timing we have almost the same cost as Neural ODE, but we have the additional cost of evolution operator. However, we emphasize that the latter cost is a lower order term. We will add timings.

5. **R3:** The central contribution hinges on the mentioned problem of neural ODEs. A: We kindly note that this is not the case. Neural ODE provides a potentially weak baseline, and so in addition to comparing to it in Tab.-R 1 above, we also compare to ANODE which is the corrected version of it. As you can see, Neural ODE results are significantly worse than ANODE/ANODEV2. All results in the paper were reported with ANODE to allow for fair comparison.

6. **R3:** The derivation of optimality for the coupled ODE is interesting but standard from dynamical systems. A: That's a fair point. We applied known methods to derive the optimality conditions. We will clarify this in the paper.

7. **R3:** Is reference 9 a peer-reviewed paper? A: Reference [9] appeared on arxiv at the time of submission of this paper, but is now listed as an accepted paper in IJCAI 2019 conference.

[Meta-Review · NeurIPS 2019]

The reviewers feel there is an interesting contribution here with a novel idea of extending the Neural ODEs setting using a coupled ODE framework. I believe the paper should be accepted.